# MMVText: A Large-Scale, Multidimensional Multilingual Dataset for Video Text Spotting

## Abstract

Video text spotting is crucial for numerous real application scenarios, but most existing video text reading benchmarks are challenging to evaluate the performance of advanced deep learning algorithms due to the limited amount of training data and tedious scenarios. To address this issue, we introduce a new large-scale benchmark dataset named **M**ultidimensional **M**ultilingual **V**ideo Text (MMVText), the first large-scale and multilingual benchmark for video text spotting in a variety of scenarios. There are mainly three features for MMVText. Firstly, we provide **510** videos with more than **1,000,000** frame images, four times larger than the existing largest dataset for text in videos. Secondly, our dataset covers 30 open categories with a wide selection of various scenarios, *e.g., life vlog, sports news, automatic drive, cartoon, etc*. Besides, caption text and scene text are separately tagged for the two different representational meanings in the video. The former represents more theme information, and the latter is the scene information. Thirdly, the MMVText provides multilingual text annotation to promote multiple cultures live and communication. In the end, a comprehensive experimental result and analysis concerning text detection, recognition, tracking, and end-to-end spotting on MMVText are provided. We also discuss the potentials of using MMVText for other video-and-text research. The dataset and code can be found at [github.com/weijiawu/MMVText](github.com/weijiawu/MMVText).

## 1   Introduction

Text reading [18, 12] has received increasing attention due to its numerous applications in computer vision, e.g., document analysis, image-based translation, image retrieval [29, 23], etc. With the advent of deep learning and abundance in digital data, reading text from images has made extraordinary progress in recent years with a lot of great public datasets [8, 13, 5] and algorithms [35, 44, 19, 17]. By contrast, video text spotting almost remains at a standstill for the lack of large-scale multidimensional practical datasets, which limited numerous applications of video text, *e.g.,* video understanding [32], video retrieval [7], video text translation, and license plate recognition [1], etc.

Most existing algorithms [44, 35, 16] in text detection and recognition deal with only static frames. Therefore, one intuitive drawback of these approaches is that they do not necessarily work well in the video domain, while at the same time they do not take advantage of the extra information present in the video (*e.g.,* tracking already detected regions). Moreover, the quality of the image is generally worse than static images, due to motion blur and out of focus issues, while video compression might create further artefacts. Due to these interferences, methods designed for still images, may fail to obtain reliable detection and recognition results when applied to a video frame. Most importantly, these methods based on image-level can not obtain text tracking information in video. However, spatio-temporal information in video is vital for a number of real-world applications. For example, video understanding and video caption translation all require temporal text information in sequential frames. There have been a few previous works [40, 38] in the community for attempting

Submitted to the 35th Conference on Neural Information Processing Systems (NeurIPS 2021) Track on Datasets and Benchmarks. Do not distribute.

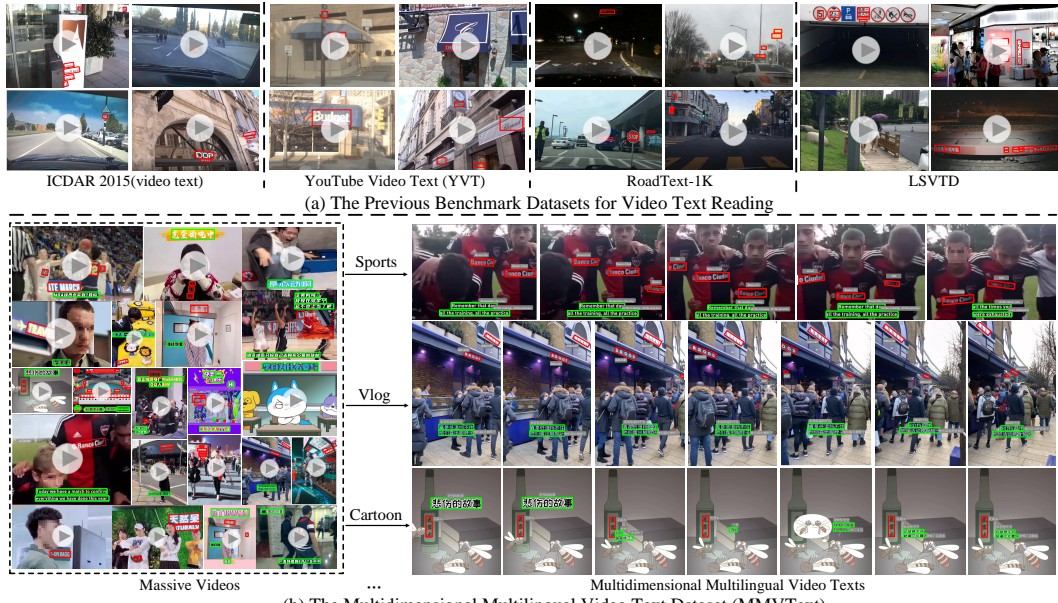

Figure 1: **Example Sequences and Annotations**. Unlike the previous benchmarks, our MMVText contains a wide variety of scenarios and multi-languages. The caption text and scene text are separately tagged for the two different representational meanings.

to develop text reading in videos, and there is a handful of datasets [25, 14] that support the research. ICDAR2015 (Text in Videos) [13], as one of the common datasets, was introduced during the ICDAR Robust Reading Competition in 2015 and mainly includes a training set of 25 videos (13,450 frames in total) and a test set of 24 videos (14,374 frames in total). The videos were categorized into seven scenarios: walking outdoors, searching for a shop in a shopping street, browsing products in a supermarket, etc. YouTube Video Text (YVT) [25] dataset harvested from YouTube, contains 30 videos (13,500 frames in total), 15 for training, and 15 for testing. The text content in the dataset can be divided into two categories, overlay text (*e.g.,* captions, songs title, logos) and scene text (*e.g.,* street signs, business signs, words on shirt). RoadText-1K [26] are sampled from BDD100K [42], includes 700 videos (210,000 frames) for training and 300 videos for testing. The texts in the dataset are all obtained from driving videos and match for driver assistance and self-driving systems. LSVTD [4] includes 100 text videos, 13 indoor (*e.g.,* bookstore, shopping mall) and 9 outdoor (*e.g.,* highway, city road) scenarios. The existing video text benchmarks are limited by the amount of training data (less than 300k frames) and tedium data scenarios, as shown in Figure. 1 (a). There are only a few outdoor scene text videos with 13k frames in ICDAR2015 (video text). Similar situation for YVT, RoadText-1k and LSVTD, the training set is limited and the dataset scenarios are tedious. This makes it difficult to evaluate the effectiveness of more advanced deep learning models.

To address this issue, our work intends to contribute a large-scale, multidimensional multilingual benchmark dataset (MMVText) to the community for developing and testing video text reading systems that can fare in a realistic setting. Our dataset has several advantages. Firstly, the large training set (*i.e.,* 1,010,848 video frames) enables the development of deep design specific for video text spotting. Secondly, MMVText is a multilingual multidimensional dataset. Abundant videos in various scenarios (*e.g.,* driving, street view, news reports, cartoon) are provided for representing real-world scenarios, as shown in Figure. 1 (b). Thirdly, caption and scene text are separately tagged for the two different representational meanings in the video. This is in favor of other tasks, such as video understanding and video retrieval. The main contributions of this work are three folds:

- We propose a large-scale, multidimensional, and multilingual video text reading benchmark named MMVText. The proposed dataset span various video scenarios, text types, multi-stage tasks and is four times the existing largest dataset.

- Caption text and scene text are separately tagged for the two different representational meanings in the video. This favors other tasks, such as video understanding, video retrieval, and video text translation.

- We evaluate the current state-of-the-art techniques for scene text detection, recognition, text tracking, and end-to-end video text spotting. Besides, a thorough analysis of performance on this dataset is provided.

# 2 Related Work

## 2.1 End-to-End Text Reading

For image-level text reading, various methods [15, 9, 19] based on deep learning have been proposed and have improved the performance considerably. Li et al. [15] proposed the first end-to-end trainable scene text spotting method. The method successfully uses a RoI Pooling [27] to joint detection and recognition features via a two-stage framework. Liao et al. [19] propose a Mask TextSpotter which subtly refines Mask R-CNN and uses character-level supervision to detect and recognize characters simultaneously. However, these methods based on the static image can not obtain temporal information in the video, which is essential for some downstream tasks such as video understanding.

Compared to text reading in a static image, video text spotting methods are rare. Yin et al. [41] provides a detailed survey, summarizes text detection, tracking and recognition methods in video and their challenges. Wang et al. [36] introduced an end-to-end text recognition method to detect and recognize text in each frame of the input video. Multi-frame text tracking is employed through associations of texts in the current frame and several previous frames to obtain final results. Cheng et al. [4] propose a video text spotting framework by only recognizing the localized text one-time. To promote text reading in the video, we attempt to establish a standardized evaluation and benchmark (MMVText), covering various open scenarios and multilingual text annotation.

## 2.2 Text Reading Datasets for Static Images

The various and practical benchmark datasets [13, 33, 14, 5] contribute to the huge success of scene text detection and recognition at the image level. ICDAR2015 [13] was provided from the ICDAR2015 Robust Reading Competition, which is commonly used for oriented scene text detection and spotting. Google glasses capture these images without taking care of position, so text in the scene can be in arbitrary orientations. ICDAR2017MLT [24] is a large-scale multilingual text dataset, which is composed of complete scene images which come from 9 languages, and text regions in this dataset can be in arbitrary orientations, so it is more diverse and challenging. ICDAR2013 [14] is a dataset proposed in the ICDAR 2013 Robust Reading Competition, which focuses on horizontal text detection and recognition in natural images. The COCO-Text dataset [33] is currently the largest dataset for scene text detection and recognition. It contains 50,000+ images for training and testing. The COCO-Text dataset is very challenging since the text in this dataset is in arbitrary orientation.

## 2.3 Text Reading Datasets for Videos

The development of video text spotting is limited in recent years due to the lack of efficient data sets. ICDAR 2015 Video [14] consists of 28 videos lasting from 10 seconds to 1 minute in indoors or outdoors scenarios. Limited videos (*i.e.,* 13 videos) used for training and 15 for testing. Minetto Dataset [22] consists of 5 videos in outdoor scenes. The frame size is 640 x 480 and all videos are used for testing. YVT [25] contains 30 videos, 15 for training and 15 for testing. Different from the above two datasets, it contains web videos except for scene videos. USTB-VidTEXT [40] with only five videos mostly contain born-digital text (captions and subtitles) sourced from Youtube. RoadText-1K provides a driving videos dataset with 1000 videos. The 10-second long video clips in the dataset are sampled from BDD100K [42]. As shown in Table. 1, the existing datasets contain a limited training set and tedium video scenarios. To promote the development of video text reading and extension of application based on video text, we create a large scale, multidimensional and multilingual dataset, and attempt to provide a more reasonable metric.

# 3 MMVText Benchmark

This section firstly introduces the collection and annotation of MMVText and provides a comprehensive analysis and comparison. And then, the related tasks and corresponding metrics are described. Finally, we discuss the link to application scenarios and potential impacts.

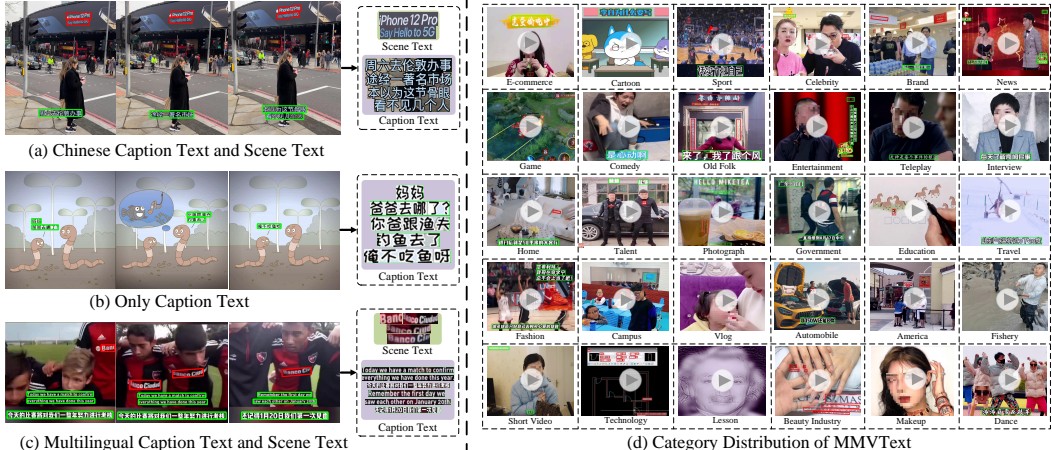

Figure 2: **Distributions of MMVText**. (a) Chinese caption and English scene text. (b) Only Chinese caption. (c) Multilingual caption and English scene text. (d) The benchmark dataset covers a wide and open range of life scenes (30 categories) with multilingual texts. Caption text (blue box) and scene text (red box) are distinguished in MMVText, which is favorable for downstream tasks.

## 3.1   Data Collection and Annotation

**Data Collection.** To obtain abundant and various text videos, we first start by acquiring a large list of text videos class using *KuaiShou*[1] - an online resource that contains billions of videos with various scene text from cartoon movies to human relation. Then, we choose 30 live video categories, *i.e.,* , *E-commerce*, *Game*, *Home*, *Fashion*, and *Technology*, as shown in Figure. 2 (d). With each raw video category, we first choose the video clips with text, then make two rounds of screening to remove the ordinary videos. As a result, we obtain 512 videos with 1,010,848 video frames, as shown in Table 1. Finally, to fair evaluation, we divide the dataset into two parts: the training set with 641,049 frames from 331 videos, and the testing set with 369,799 frames from 179 videos. As shown in Figure 2 (a), different from the existing data sets, which only focus on one type of video text and the video scene is limited, our dataset not only care about scene text reading in the real world, but also focus on caption texts in the video. For the most part, caption text represents more global information than scene text, which is quite favorable for some downstream tasks, *e.g., video understanding, video caption translation.* Therefore, the MMVText can cover a wider and open range of life scenes, and contains various text with a more comprehensive description of the video.

**Data Annotation.** We invite a professional annotation team to label each video text with four kinds of description information: the bounding box describing the location information, judging the tracking identification (ID) of the same text instance, identifying the content of the text information, and distinguishing the category label of the caption or scene text. To save the annotation cost, we first sample the videos, annotate each sampled video frame at an instance level, and then transform the annotation information from the sampled video frame to the unlabeled video frame by interpolation. *For video sampling*, we use uniform sampling with a sampling frequency of 7 to sample all the videos in the dataset, and obtain the sampled video frame set. *For sampling video frame annotation*, each text instance is labeled in the same quadrilateral way as in the ICDAR 2015 incidental text dataset [45]. In addition, the text instance also will be marked with two description information: the category of the caption or scene, and the recognition content. After the spatial location, content, and category of the video text are determined, the annotator will determine the tracking ID by browsing the length of the same video text in the continuous sampling video frames. We also invited other text-related people to conduct two rounds of cross-checking to ensure the annotation quality. *For video frame recovery*, each text instance is marked with tracking ID and recognition content, so we can judge whether different texts in adjacent sampling frames are the same text. After determining the same text instance, we first determine whether the text annotation of the sampled video frame is the starting and end frame of the text instance. If not, we look forward and backward for the starting and end position of the text instance and label it. Then we use the linear interpolation way to calculate the position of the text object in the middle of the unmarked video frame, and give tracking ID, recognition content,

---

[1]https://www.kuaishou.com/en

Table 1: **Statistical Comparison.** Comparisons between MMVText and existing datasets for caption and scene text in videos. *D, T, and S* denotes the Detection, Tracking, and Spotting respectively.

| Dataset | Category | MLingual | Scenario | Videos | Frames | Texts | Task |
|---|---|---|---|---|---|---|---|
| AcTiV-D [43] | Caption | - | News video | 8 | 1,843 | 5,133 | D |
| UCAS-STLData [3] | Caption | - | Teleplay video | 3 | 57,070 | 41,195 | D |
| USTB-VidTEXT [40] | Caption | - | Web video | 5 | 27,670 | 41,932 | D&S |
| YVT [25] | Scene | - | Incidental | 30 | 13,500 | 16,620 | D&T&S |
| ICDAR 2015 VT [45] | Scene | - | Incidental | 51 | 27,824 | 143,588 | D&T&S |
| LSVTD [4] | Scene | ✓ | Incidental | 100 | 66,700 | 569,300 | D&T&S |
| RoadText-1K [26] | Scene | - | Driving | 1000 | 300,000 | 1,280,613 | D&T&S |
| MMVText (ours) | Both | ✓ | Open | 510 | **1,010,848** | **4,513,525** | D&T&S |

154 and category. After all the video annotations are restored, we carry out another round of double
155 detection correction. As a labor-intensive job, the labeling process takes 30 men in two months, *i.e.,*
156 20,160 man-hours, to complete about 200,000 sampled video frame annotations.

## 3.2 Dataset Analysis

157

158 **Statistic Comparison.** The qualitative and statistic comparison between the established MMVText
159 and other datasets are visualized in Figure. 1, and summarized in Table. 1. *Category* denotes the
160 category of the text type in the corresponding dataset. *MLingual* denotes whether the dataset contains
161 multiple language texts. *Scenario* denotes the scene range of the video. *Videos, Frames, Texts*
162 represents the number of videos, video frames, video texts in the dataset, respectively. *Task* denotes
163 which tasks the dataset supports. **Caption Text and Scene Text.** For comprehensive evaluation and
164 research, we not only expand the scale of the dataset (*i.e.,* , the number of videos, video frame, and
165 video text), and label the spatial quadrilateral position, recognition content, and tracking ID, but also
166 additionally collect and annotate the category of caption or scene for each text instance. As shown
167 in Figure. 2 (a), in a video, different types of text instances may exist simultaneously, and they are
168 helpful to understand videos synergistically. Concretely, caption text can directly show the dialogue
169 between people in video scenes and represent the time or topic of the video scenes, scene text can
170 unambiguously define the object and can identify important localization and road paths in video
171 scenes. Besides, nowadays, caption text frequently exists in all kinds of life scenarios video. Even
172 for some videos, without any scene texts, there is a lot of caption text, as shown in Figure. 2 (b). To
173 favor downstream tasks (*e.g.,* video text translation, video understanding, and video retrieval), we
174 also provide multilingual text annotations, as shown in Figure. 2 (c).

175 To provide the community with unified text-level quantitative descriptions, and facilitate controlled
176 evaluation for different approaches, we will compare our dataset with caption or scene text datasets
177 from four aspects, *i.e.,* text description, video scene, dataset size, and supported tasks. *For text*
178 *description attribute* (*i.e., Category, MLingual*), our MMVText contains both types (caption and
179 scene) of video text and has multi-language features, which obviously has more extensive description
180 ability than caption or scene text dataset. *For video scene attribute* (*i.e., Scenario*), the caption
181 text datasets choose videos with certain professional purposes (*e.g.,* news reports, TV dramas, and
182 documentaries), which shows that the scenes they cover are relatively limited. And the existing
183 scene text datasets often choose some video scenes captured by mobile shooting, and the number of
184 collectors is small, the range of captured scenes is also limited. However, the videos in our dataset
185 are from videos uploaded voluntarily by all kinds of users. Therefore, the proposed MMVText
186 covers various scenarios, but it also brings significant challenges to researchers. *For the size of*
187 *the dataset*(*i.e., Videos, Frames, Texts*), we can find that our MMVText has advantages over the
188 superimposed caption text dataset and the scene text dataset in the indicators of videos, frames, and
189 texts. The number of videos in RoadText-1K is more than ours (*1,000 vs. 510*), but the number of
190 frames in RoadText-1K is far less than ours (*300,000 vs. 1,010,848*), which imply that the average
191 video length of RoadText-1K is much shorter than ours (*300 vs. 1,982*). *For the supported tasks*,
192 the proposed MMVText supports four common video text tasks: detection, recognition, video text
193 tracking, end to end video text spotting. The focus and application scenarios of each task is entirely
194 different. For example, detection task used in the static image focus on localization performance,
195 paving the way for recognition task, which apply to license plate recognition. End to end video text
196 spotting task focuses on recognition and tracking performance, which apply to video understanding

and video retrieval. In conclusion, the high efficiency of MMVText for evaluating advanced deep learning methods is very favorable for promoting various text reading applications in real life.

## 3.3 MMVText Tasks and Metrics

Standardized benchmark metrics are crucial as same as the dataset for the majority of computer vision applications, and we attempt to provide a reasonable evaluation for video text reading methods. The proposed MMVText mainly includes two tasks: (1) Video Text Tracking, aimed at describing text location information in continuous frames. (2) End to End Text Spotting in Videos, to understand text and track multiple frames. For the detection and recognition task, we also provide corresponding experimental results and analysis in the experiments.

Most tracking tasks all use the $MOT$ metrics [2], which was launched to establish a standardized evaluation of multiple object tracking methods. The same case for video text tracking, the ICDAR2013 Robust Reading Challenge [14] for video text reading adopts $MOTP$ (Multiple Object Tracking Precision) and $MOTA$ (Multiple Object Tracking Accuracy) as the metrics. Following the previous works [14, 26], MMVText evaluates text tracking methods in video and compares their performance with the MOTA and MOTP. Besides, $ID_{F1}$ as the new metrics for tracking is presented from some tracking works [6, 28] in recent year. $ID_{F1}$ is the ratio of correctly identified detections over the average number of ground-truth and computed detections. And the metric is more reasonable to evaluate ID switches in some cases. We also evaluate the metrics in MMVText by:

$$ID_{F1} = \frac{2ID_{tp}}{2ID_{tp} + ID_{fp} + ID_{fn}}, \tag{1}$$

where $ID_{tp}$, $ID_{fp}$ and $ID_{fn}$ refer to true positive, false positive and false negative of matching ID. Besides, the ID metric [6] also includes $MT$ (Mostly Tracked) Number of objects tracked for at least 80 percent of lifespan, $ML$ (Mostly Lost) Number of objects tracked less than 20 percent of lifespan.

In Task2 (End to End Text Spotting in Videos), the objective of this task is to recognize words in the video as well as localize them in terms of time and space. And we argue that the final recognition result is more important than text localization in videos. Thus, we modify the $ID_{F1}$ to $TID_{F1}$, which focuses on text instance ID tracking and recognition results that be required by many downstream tasks. More specifically,

$$TID_{tp} = \sum_{h} \sum_{t} m(h, o, \triangle_t, \triangle_s, \triangle_r), \tag{2}$$

$$TID_{F1} = \frac{2TID_{tp}}{2TID_{tp} + TID_{fp} + TID_{fn}}, \tag{3}$$

where $\triangle_t$, $\triangle_s$ and $\triangle_r$ refer to time matching, space location matching and recognition result matching. And $h$ and $o$ denote hypothesis and true text trajectory with recognition result. The match of $h$ and $o$ is a true positives of text ID (*i.e.*, $TID_{tp}$) when these conditions (*i.e.*, $\triangle_t$, $\triangle_s$ and $\triangle_r$) are met. Similarly, false positive (*i.e.*, $TID_{fp}$) and false negative (*i.e.*, $TID_{fn}$) of text ID can be obtained for $TID_{F1}$ calculation. More details concerning metrics in supplarmentary material.

## 3.4 Methods

Text detection and recognition in the static image have made tremendous progress, and abundant great work [35, 44, 30] be proposed. By contrast, the counterparts in video text reading are rare and lack quality open-source algorithms. Therefore, we adopt various mature techniques in the static image to better evaluate the efficiency of MMVText.

**Detection**. The deep learning-based text detection methods can be roughly divided into two categories: regression-based method and segmentation-based method. EAST [44] as one of the popular regression-based methods is used to test our MMVtext. The method adopts FCNs to predict shrinkable text score maps, rotation angles and perform per-pixel regression, followed by a post-processing NMS. For segmentation based methods, we adopt PSENet [35] and DB [16] to evaluate our MMVtext. PSENet [35] generates various scales of shrinked text segmentation maps, then gradually expands kernels to generate the final instance segmentation map. Similarly, DB [16] utilizes the shrinked

text segmentation maps and differentiable binarization to detect text instances. **Recognition**. Recent methods mainly use two techniques to train the scene text recognition model, namely Connectionist Temporal Classification (CTC) and attention mechanism. In CTC-based methods, CRNN [30] as the representation, which introduced CTC decoder into scene text recognition with a Bidirectional Long Short-Term Memory (BiLSTM) to model the feature sequence. In Attention-based methods, RARE [31] firstly normalizes the input text image using the Spatial Transformer Network (STN [11]), then utilizes CNN to extract feature and captures the contextual information within a sequence of characters. Finally, it estimates the output character sequence from the identified features with the attention module.

**Text Tracking Trajectory Generation**. With text detection and recognition in a static image, we only obtain text localization and recognition information without temporal information, which are insufficient for video spotting evaluation (*e.g.,* $TID_{F1}, MOTA$ and $MOTP$). The work [36] based on multi-frame tracking provides a method to track text instances temporally based on attributes of the text objects in multiple frames. Following the work [36], we link and match text objects in the current frame and several frames by IOU and edit distance of text.

### 3.5 Link to Real Applications

Text understanding in static images has numerous application scenarios: (1) Automatic data entry. SF-Express [2] utilizes OCR techniques to accelerate the data entry process. NEBO [3] performs instant transcription as the user writes down notes. (2) Autonomous vehicle [21, 20]. Text-embedded panels carry important information, *e.g.,* geo-location, current traffic condition, navigation, and etc. Similarly, there are many application demands for video text understanding across various industries and in our daily lives. We list the most outstanding ones that significantly impact, improving our productivity and life quality. Firstly, automatically describing video with natural language [39, 37] can bridge video and language. Secondly, video text automatic translation [4] can be extremely helpful as people travel, and help video-sharing websites [5] to cut down language barriers. More details and analyses for application scenarios concerning MMVText in the supplementary material.

## 4 Experimental

In this section, we conduct experiments on our MMVText to demonstrate the effectiveness of the proposed benchmark. Note that we denote Ground Truth of ID tracking in all the experiments, Mostly Tracked and Mostly Lost as 'GT', 'MT' and 'ML', respectively.

### 4.1 Implementation Details

All of the experiments use the same strategy: (1) Training detector and recognizer with MMVText. (2) Matching text objects with corresponding text tracking trajectory id. *Detection*: without pretrained model, we train detectors directly with training set (*i.e.,* 641,049 frame images) of MMVText. *Recognition*: the network is pre-trained on the *chinese ocr*[6] and MJSynth [10], and further fine-tuned on our MMVText. All of our experiments are conducted on 8 V100 GPUs. PSENet [35], EAST [44] and DB [16] are adopted as the base detectors because of their popularity. CRNN [30] and RARE [31] as the base text recognizers to evaluate our MMVText. In the PSENet, EAST, DB, CRNN and RARE experiments, all settings follow the original reports.

### 4.2 Attribute Experiments Analysis

**Text Tracking in Different Scenarios.** Figure. 3 (a) gives the tracking performance $ID_{F1}$ of EAST [44] in different scenarios of MMVText. The model achieves the best performance with a $ID_{F1}$ of 57% in cartoon videos, since the conspicuous text instances and simple background are

---

[2]https://www.sf-express.com/cn/sc/

[3]https://www.myscript.com/nebo/

[4]https://translate.google.com/intl/en/about/

[5]https://www.youtube.com/

[6]https://github.com/YCG09/chinese_ocr

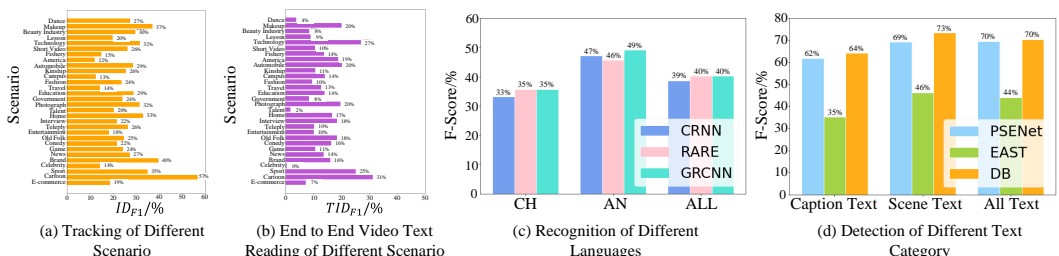

| (a) Tracking of Different Scenario | (b) End to End Video Text Reading of Different Scenario | (c) Recognition of Different Languages | (d) Detection of Different Text Category |

Figure 3: **Attribute Experiments of MMVText**. (a) Tracking performance (*i.e.,* $ID_{F1}$) with EAST [44] in different scenarios. (b) End to end video text spotting performance (*i.e.,* $TID_{F1}$) with PSENet [35] and CRNN [30] in different scenarios. (c) Recognition accuracy of different models in different languages. (d) Detection performance of different model in caption or scene text. 'CH', 'AN' and 'ALL' refer to 'Chinese Characters', 'Alphanumeric Characters' and 'All Characters'.

Table 2: **Detection and Recognition Performance on MMVText**. Frame level text Detection and Recognition performance of existing models on MMVText. 'CH', 'AN' and 'ALL' refer to 'Chinese Characters', 'Alphanumeric Characters' and 'All Characters'.

| Detection Performance/% | | | | Recognition Performance/% | | | | | | |
|---|---|---|---|---|---|---|---|---|---|---|
| Method | Precision | Recall | F-score | Method | Pretrained | | | Fine tuned | | |
| | | | | | CH | AN | ALL | CH | AN | ALL |
| EAST [44] | 52.2 | 38.1 | 44.1 | CRNN [30] | 26.0 | 32.1 | 23.2 | 33.2 | 47.1 | 38.6 |
| PSENet [35] | 74.3 | 65.2 | 69.5 | RARE [31] | 25.2 | 34.2 | 23.5 | 35.6 | 45.7 | 40.2 |
| DB [16] | 77.2 | 64.5 | 70.3 | GRCNN [34] | 23.1 | 39.8 | 26.7 | 35.6 | 49.2 | 40.3 |

designed in cartoon videos. By comparison, several scene categories obtain extremely dissatisfied performance due to complex background and various text appearance, such as $Campus$ and $Travel$.

**End to End Text Spotting in Different Scenarios.** Figure. 3 (b) gives the end-to-end performance $TID_{F1}$ using PSENet [35] and CRNN [30] in different scenarios of MMVText. Similar to tracking performance using EAST [44], the end-to-end video spotting performance shows the best performance with a $TID_{F1}$ of 31% in scenario of $Cartoon$.

**Text Recognition for Different Language.** As shown in Figure. 3 (c), the text recognition results for different languages are provided. In summary, the alphanumeric recognition result (about 47%) is better than the Chinese recognition result (about 35%), regardless of the models. The final results (about 40%) for all characters are satisfactory, can not meet the requirement of the application.

**Text Detection for Different Text Category.** As shown in Figure. 3 (d), we provide the detection performance comparison for different models in different text categories (*i.e.,* caption text or scene text) of MMVText. It is obvious that the performance for scene text is better than the counterpart of caption text, regardless of which detection model. The prime reason is that caption texts are all long text, a different case to detect without any model refinement.

### 4.3 Text Detection and Recognition in Images

Although text detection and recognition in static images are not the focus in this work, we provide the corresponding performance for comparison, as shown in Table. 2. For text detection, we adopts EAST [44], PSENet [35] and DB [16] to evaluate the proposed MMVText. We observe that frame-level text detection and recognition results on MMVText are not unsatisfactory, with lower results than these methods report on existing scene text datasets. For example, EAST only obtains an f-score of 44.1% compared to the F-score of 80.7% on icdar2015 [45]. For text recognition, CRNN [30] based on CTC loss, RARE [31] with attention mechanism and GRCNN [34] as the base text recognizers to test our MMVText. The text annotation in our MMVText covers two languages (*i.e.,* English and Chinese), thus we conduct several experiments for each language. 'CH' and 'AN' refer to Chinese text instances and alphanumeric characters. 'ALL' denotes all characters regardless of which language. Similar to the detection task, the recognition model only yields about 40% accuracy on our dataset, but the same model reports $> 90+$ on most benchmark datasets [14] for scene text recognition. The main reasons have two points: (1) The proposed MMVText is multilingual, and the category number

Table 3: **Text Tracking Performance on MMVText.** Text tracking trajectory id generation use a method proposed in [36].

| Method | MOTP | MOTA | $ID_P$/% | $ID_R$/% | $ID_{F1}$/% | GT | MT | ML |
|---|---|---|---|---|---|---|---|---|
| EAST [44] | 0.275 | -0.301 | 23.5 | 22.9 | 23.2 | 48321 | 9680 | 35802 |
| PSENet [35] | 0.112 | 0.334 | 34.7 | 26.7 | 29.9 | 48321 | 12755 | 33410 |
| DB [16] | 0.102 | 0.438 | 33.7 | 29.9 | **31.7** | 48321 | 14958 | 31444 |

Table 4: **End to End Video Text Spotting Performance on MMVText.** Text tracking trajectory id generation use a method proposed in [36]. $TID_P$, $TID_R$, $TID_{F1}$, $MOTP_T$ and $MOTA_T$ refer to the corresponding metrics with recognition results in Table. 3.

| Method | | $TID_P$/% | $TID_R$/% | $TID_{F1}$/% | $MOTA_T$ | $MOTP_T$ | MT | ML |
|---|---|---|---|---|---|---|---|---|
| Detection | Recognition | | | | | | | |
| EAST [44] | CRNN [30] | 5.3 | 5.1 | 5.2 | -0.835 | 0.173 | 1564 | 45963 |
| | RARE [31] | 3.0 | 3.6 | 3.2 | -1.130 | 0.173 | 1265 | 47104 |
| PSENet [35] | CRNN [30] | 14.7 | 9.8 | 11.8 | -0.300 | 0.197 | 3790 | 42957 |
| | RARE [31] | 15.2 | 10.4 | 12.4 | -0.280 | 0.201 | 3821 | 42417 |
| DB [16] | CRNN [30] | 15.6 | 9.6 | 11.9 | -0.284 | 0.230 | 3356 | 43246 |
| | RARE [31] | 20.1 | 15.2 | **17.3** | -0.293 | 0.150 | 4230 | 39650 |

of Chinese characters in real-world images is much larger than those of Latin languages. (2) The video texts are quite blurred, out-of-focus, and the distribution of characters is relatively smaller than the static image counterparts.

### 4.4 Text Tracking and Spotting in Videos

**Video Text Tracking.** Table. 3 shows the comparing results of text tracking on MMVText. We observe that the overall performances of the used detectors are dissatisfactory on MMVText. Besides, the $IDF_1$ of EAST [44] is lower with $6.7\%$ gap than that of PSENet [35]. The main reason is that MMVTtext contains a mass of long text instances, but regression-based EAST can not deal with the long text cases well. The performance of DB is similar to that of PSENet for both all are the segmentation-based methods. According to Table. 3, $MOTP$ shows a better performance than $MOTA$. We argue that detectors such as PSENet or DB provide strong detecting capacity, but the tracking ability is relatively weak. By comparison, $IDF_1$ is a comprehensive metric for object ID tracking. $ID_{F1}$ ($31.7\%$) of DB achieves the best performance of the three detectors, and EAST shows the worst performance with a $ID_{F1}$ of $23.2\%$.

**End to End Text Spotting in Video**. Detection or text tracking tasks are paving the way for the recognition task. Table. 4 shows the performance of text spotting in the video. And $TID_{F1}$ in Equation. 3 as an integrated metric to evaluate algorithms in spatial location, content, and temporal information three dimensions. Similar to the text tracking performance of EAST, the corresponding performance $TID_{F1}$ using CRNN [30] as the recognizer in video text spotting is still not satisfied with a $5.2\%$ $TID_{F1}$. The combination of DB [16] and RARE [31] achieves the best performance with a $17.3\%$ $TID_{F1}$ among all the cases, but the performance still is inadequate to meet application requirements. MT (Mostly Tracked) and ML (Mostly Lost) as the metrics concerning statistical number can be used to evaluate from another aspect. For the combination of DB [16] and RARE [31], 39650 text tracking trajectories are lost, less than 20 percent of lifespan. By comparison, only 4230 tracking trajectories are satisfactory, more than 80 percent of lifespan tracked.

## 5 Conclusion and Future Work

In this paper, we establish a large-scale multidimensional and multilingual dataset for video text tracking and spotting, termed as MMVText, with four description information, *i.e.,* , bounding box, tracking ID, recognition content, and text category label. Compare with the existing benchmarks, the proposed MMVText mainly contains three advantages: large-scale, multidimensional, multilingual. MMVText spans various video scenarios, text types, and multi-stage tasks, promoting video text research. We also conduct several experiments on this dataset and shed light on what attributes are especially difficult for the current task, which cast new insight into the video text tracking, spotting field. In general, we hope the MMVText would facilitate the advance of video-and-text research.

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
