# OpenReview forum: "MMVText: A Large-Scale, Multidimensional Multilingual Dataset for Video Text Spotting"
_NeurIPS.cc/2021/Track/Datasets_and_Benchmarks/Round1 — Submitted to NeurIPS 2021 Datasets and Benchmarks Track (Round 1)_

### Official Review · Reviewer_qYEt · 2021-07-02
**A new large (1 million frames) and diverse (30 categories) multilingual (mostly Chinese/English) benchmark for text detection/recognition in video.**

**Rating:** 6
**Confidence:** 4

**Strengths:**

The proposed MMVTest dataset is potentially significant/impactful for the community working on detecting/recognizing text in images/video, as it offers a larger/more diverse benchmark. It appears comprehensively labeled: texts are annotated with bounding boxes, ID information, textual content and category label (caption or scene text).

The dataset may also be of interest to those working on other video-related tasks (video understanding, video description etc) and has several other application scenarios (e.g. automatic video text translation), which however may not be strictly specific to this dataset, but to this research direction in general.

The evaluation supports several tasks, including text detection/recognition in individual images as well as Video Text Tracking (temporal localization and re-ID) and End to End Text Spotting in Videos (focus on recognition)

The authors promise to share the data and code. The given URL (https://github.com/weijiawu/MMVText-Benchmark) currently still has several TBDs. The project is open source under BSD-3 license.

**Weaknesses:**

Based on the weaknesses below, I recommend that the authors revise the paper after addressing the comments below.

= No detailed dataset statistics
The authors emphasize the diversity of the proposed dataset, however they do not provide a breakdown of how many videos/frames there are for each of the 30 categories.
Also, it would be helpful to include the text statistics broken down by language (Chinese, English, etc) as well as label type (scene/caption).
This is *a dataset paper*, so all such details should be included (at least in the supplemental).

=Contribution/Comparison to prior datasets
While the authors discuss prior "video w/ text" datasets, it remains unclear how the proposed dataset complements them or whether it is indeed more challenging (L186). Some things the authors could have included, are e.g. discussion of dataset difficulty (prior vs. theirs) as well as to show that a model trained on the proposed large/diverse dataset transfers to prior smaller datasets. That would help demonstrate the dataset usefulness and complementarity to prior works.
I also wonder, can one combine all the existing datasets together? Will that not lead to the largest/most diverse training data?
Besides (Sec 3.1), the authors start their search on a video platform, with *billions* of video, to collect ~500 videos in total. I found this number a little underwhelming. While the #frames is x3-4 times larger than in prior wok, it is not like an order of magnitude or more increase in size. Also it is a little unclear what the main bottleneck is in scaling this further, is it the manual annotation process?

=Chosen methods
The authors mostly focus on methods designed for static images, not video (with one exception). One argument for that is that no video-based detection/recognition methods have publicly available code. While I can see that this is an issue, it still seems not adequate to just focus on image-based methods. Moreover, the author make comparisons to how these methods performed on image-based tasks, which does not provide much insight, i.e. it would be more interesting to know whether the proposed dataset is more or less difficult that other existing *video-based* datasets (as discussed above).

=Limited takeaways
Sec 4.2 is supposed to provide some insights into the challenges of the proposed dataset. However, the authors provide little insights, e.g. what makes some of the video types (e.g. Campus, Travel, Celebrity) more difficult than others? Is it possible that these are simply less frequent? (This is where the dataset stats would help!) Or are there any specific features that make them harder? Perhaps some have more caption text and others have more scene text? An additional breakdown may be interesting! Fig 3 (c) and Table 2 could have included a more detailed language breakdown, e.g. include English specifically, especially since the authors argue that Chinese is more difficult to recognize (L311-312). Overall, more insights could be included!

=Paper presentation/clarity
Paper has multiple issues with presentation and clarity, writing has many issues, more on that below.

**Additional Feedback:**

Sec 4: I would not start experimental results with Sec 4.2 (Attribute Experiments Analysis) but rather with the core 4 tasks, and later discuss this.

I recommend to illustrate MT/ML in %, no absolute numbers.

Sec 3.5 seems out of place where it is. Could go to conclusion/discussion?

In the Checklist, to the question "If you are including theoretical results..." you should probably say N/A, as this is not a theoretical paper.

**Clarity:**

The authors do not offer a very good motivation for the need of a new benchmark. One may guess that dataset size is one issue, but it is not clear what the other issues are.  Starting with an abstract's first sentence (which is hard to understand), what are these "tedious scenarios" that the authors keep referring to?

Are there established terms for how different tasks in this work are typically referred to? E.g. we see "text reading" (does it mean OCR?), "text spotting", etc, the task names should be standardized and the tasks should be clearly defined. Table 1 mentions 3 tasks "Detection, Tracking, and Spotting", later in text the authors mention 4 tasks (L192). Later the paper, there are inconsistent names used, e.g. "End to End Video Text Reading" vs. "End to End Text Spotting".

L14 "promote multiple cultures live and communication" : unclear

L30-36: The authors argue for challenges of text detection/recognition in video as opposed to images, are there any references to cite to support this discussion?

L43 according to this, YouTube Video Text (YVT) also has two types of text annotations, captions and scene test. However, in Table 1 it is shown as only "Scene" text.

L54 "This makes it difficult to evaluate the effectiveness of more advanced deep learning models." - why? have people tried? have they failed? do you have references?

L64-66 odd to refer tho the dataset as "multidimensional", not sure what the authors mean; also what are the "multi-stage tasks"?

L147 what is "video frame recovery"?

L154 what is "double detection correction"?

Table 1: What is the difference between "Incidental" and "Open" scenario? I think both are open, so perhaps include #categories?

L 195, 203 Is "End to end video text spotting task' an existing task or a new task? Initially it seems to be an existing task, hence the authors discuss that it is "supported" by other datasets, but then they introduce *a new metric* specific to the task, suggesting it is new?

L197 "In conclusion, the high efficiency of MMVText for evaluating advanced deep learning methods is very favorable for promoting various text reading applications in real life." : the authors can not make such a claim before they showed any such results.

L202: strange to say that the dataset "mainly includes two tasks" after claiming that it supports 3 (or 4) tasks? and in the end reporting results of 4 tasks?!

L223: it is unclear how the time matching, space location matching and recognition result matching are defined

L227 supplarmentary => supplementary

Fig 3 (b) is supposed to show two methods (PSENet and CRNN) but only one plot is present.

L313 "distribution of characters is relatively smaller than the static image counterparts." - unclear

Some points are being repeated multiple times in the paper (e.g. about the properties of the proposed dataset), making a lot of content seem redundant (e.g. Sec 3.2 "echoes" the previous sections).

**Correctness:**

Little information is given about the exact instructions given to the annotators and the annotation workflow sounds somewhat confusing (Sec 3.1). Some example annotations would be great to include, as well as the annotation interface. (Could be part of the supplemental.)

The annotators only label a subset of frames, while for the remaining frames the annotations are interpolated. We do not know how good the quality of the interpolation is. What if an important transition occurs in between (new text appears, than disappears), that would not be captured, right?

Regarding the new proposed metric (Sec 3.3), L223: it is unclear how the "time matching, space location matching and recognition result matching" are defined. The code (if provided) may help clarify that.

To the Checklist question "Did you report error bars (e.g., with respect to the random seed after running experiments multiple times)?" the answer was given Yes, but this is not accurate.

**Documentation:**

The authors promise to share the data and code. The repository under the given URL (https://github.com/weijiawu/MMVText-Benchmark) is still under construction and has has several TBDs (i.e. links to data and some code are missing).
The project is open source under BSD-3 license.

**Ethics:**

To the checklist question "Did you discuss any potential negative societal impacts of your work?" the answer was given No.
The authors could think about it and add some comments to the conclusion/discussion section in the future revision.

To the checklist question "Did you discuss whether the data you are using/curating contains personally identifiable
information or offensive content?" the answer was given "Yes, We have blurred identifiable information or offensive content." Would be interesting to learn more about how that was accomplished.

To the checklist question "Did you include the estimated hourly wage paid to participants and the total amount spent on participant compensation?" the answer was given "Yes, We have paid salary to the related participants." but that does not answer the question.

**Relation To Prior Work:**

The authors mention a survey for video w/ text methods [41] L82, however they only discuss two methods, [36] and [4]. I assume there are more than two methods in the survey?

Some additional references may be the new Vision&Language tasks that include scene text:
[a] Singh, Amanpreet, et al. "Towards vqa models that can read." Proceedings of the IEEE/CVF Conference on Computer Vision and Pattern Recognition. 2019.
[b] Sidorov, Oleksii, et al. "Textcaps: a dataset for image captioning with reading comprehension." European Conference on Computer Vision. Springer, Cham, 2020.



**Summary And Contributions:**

The paper introduces MMVTest, a new benchmark for text detection/recognition (also referred to as “spotting”) in video. Key characteristics include:
- Size (1 million frames), while the largest prior work has 300k frames; over 4 million text annotations (largest prior work has over 1 million).
- Open domain (30 different video categories, over 500 videos in total), while most prior work either specializes in a single domain or has a somewhat limited diversity of data [this is somewhat difficult to verify, more on that below]
- Multilingual, primarily Chinese and English texts; only one prior work is also multilingual, LSVTD [4]
- Separate annotations for scene text and caption text, prior works typically do one or another.
- The authors also seem to introduce a new evaluation scheme for End to End Text Spotting [more on that below]
- The authors evaluate several existing methods (mostly designed for *static* text detection/recognition) on their video benchmark.

= UPDATE after the author response / discussion period =

I have updated my rating to 6 (Marginally above acceptance threshold). My rationale is below.
Overall, I buy the authors' arguments about the unique features of the proposed dataset. The paper originally had many issues with presentation/clarity, which the authors tried to address during the discussion period.
I maintain my opinion that the authors should not claim that this new benchmark is "more challenging" than prior work as they do not show that empirically; nor do they explore its complementarity to prior datasets, as I have detailed in my review. Overall makes it challenging to draw conclusions about any comparison to prior work.
PS One of the reviewers raised a criticism about not showing the usefulness of the proposed dataset to some other downstream tasks.  While this would have been a great addition, I do believe this is out of scope for the given paper, thus I disagree with this criticism.

---

> ### Author Response · Authors · 2021-07-11
> **Response to Reviewer qYEt (Part1)**
>
> We sincerely thank the reviewer for the detailed and valuable comments/suggestions. We firmly believe that these suggestions will surely help a lot for the paper.
>
> **Q1:** *Lacking detailed dataset statistics.*
> **A1:** Thank you for the valuable suggestions. We have been provided more detailed information in the revised version:
> - **The data distribution for 30 open scenarios**: https://github.com/weijiawu/MMVText-Benchmark/blob/main/Dataset/image/Data%20Distribution.png
> - **The data distribution for text language and category**: https://github.com/weijiawu/MMVText-Benchmark/blob/main/Dataset/image/languageAndCategory1.png
>
> **Q2:** Contribution/Comparison to prior datasets. **A2:**  We want to further state the contributions of MMVText:
> - **Size**. MMVText is **15 times** larger than the existing largest dataset (*i.e.,* LSVTD[2]) with various scenarios(1,010,848 *v.s*  66,700 video frames). RoadText1k[1] contains 300k videos frames, but it only supports driving video scenarios.
> - **Supported Language.** As we know, the proposed MMVText is the first large-scale multilingual video text benchmark.
> - **Video Scenarios.** Unlike the existing datasets, MMVText provides various open scenarios, including many **new scenarios** such as Sportscast(NBA, FIFA World Cup...), Life Vlog, Game, etc. To further present the scenarios diversity of MMVText, we summarized detailed statistics in the [Table](https://github.com/weijiawu/MMVText-Benchmark/blob/main/Dataset/image/Statistical%20Comparison1.png).
> - **Annotation.** Caption text(songs title, logos) and scene text(street signs) are separately tagged for the two different representational meanings. Besides, MMVText supports rotate box annotation(four points), but many existing datasets(RoadText-1k, YVT) only support bounding box annotation(two points).
> - **Data Quality.** Videos in MMVText support high resolution (*i.e.,* 1080p), but most existing datasets(RoadText-1k, YVT)  only support 720p.
>
> Three new insights from the experiments:
> - **New Scenarios, New Challenge.** The first proposed new scenarios, *e.g., Campus, Celebrity* present unsatisfactory results than others, as shown in Figure 3 (a)-(b) and Attribute Experiments. The low performance is mainly caused by blur and out-of-focus and complex themes in the new scenarios.  The comparison for *Cartoon* and *Celebrity* can be found in https://www.youtube.com/watch?v=_Yykx0qgJZ4. (Green and blue boxes denote annotation and prediction of PSENet, respectively. )
> - **Multilingual Recognition, New Challenge.** Unlike English, Chinese contains thousands of characters(**3,856 Chinese characters v.s. 26 English characters on MMVText**), which are difficult to recognize. We also present some bad cases in https://github.com/weijiawu/MMVText-Benchmark/blob/main/Dataset/image/recognition1.png.  Table 2 and Figure 3 (c) in the Attribute Experiments all prove the phenomenon. MMVText, as *the first large-scale multilingual dataset*, **15 times** larger than LSVTD[2], can promote multilingual video text development.
> - **Long Caption Text, New Challenge.** Figure 3 (d) shows the performance of scene text is better than the counterpart of caption. The prime reason is that captions are all long text (**average width-height ratio: 6.8 for caption *v.s.* 2.3 for scene text**), the difficult cases to detect. The visualization concerning the bad case for long caption text can be found in https://www.youtube.com/watch?v=MhPuxW_bdos. (Green and blue boxes denote annotation and prediction of EAST, respectively. )
>
> **Conclusion.** Most existing video text datasets are proposed before **2019 years**, and the download links of YVT even have become invalid. We try to establish a practical standardized benchmark, MMVText, which would facilitate the advance of video-and-text research.
>
> **Q3:** *Can one combine all the existing datasets together? Will that not lead to the largest/most diverse data?* **A3:** Great question! But we still want to highlight some points:
> - **The combining one is not the largest.** As we show, the one combining all the existing datasets only provides 494,607 video frames with 2,098,381 text instances. But MMVText can provide 1,010,848 video frames with 4,513,525 text instances, **2 times larger** than the combined one.
> - **The combining one is not the most diverse.** MMVText provides various scenarios, many of them, *e.g., Game(PUBG mobile…), Sport(NBA…), News Report, TV...*, which are not supported by the existing datasets. The statistics comparison can be found in https://github.com/weijiawu/MMVText-Benchmark/blob/main/Dataset/image/Statistical%20Comparison1.png.
> - **Copyright issue.** We do not sure if there is a copyright issue.
> -  **Annotation disagreement.** For example, Roadtext-1k[1] is annotated in line-level, but LSVTD[2] provides word-level annotation. Besides, bounding box annotation(two point) on Roadtext-1k is also different with rotated rectangle box annotation(four points) on LSVTD[2].

---

> ### Author Response · Authors · 2021-07-11
> **Response to Reviewer qYEt (Part2)**
>
> We sincerely thank the reviewer for the detailed and valuable comments/suggestions. We firmly believe that these suggestions will surely help a lot for the paper.
>
> **Q4:** *What the main bottleneck is in scaling size this further?* **A4:** We argue that the main bottleneck is the huge manual annotation cost just as you indicated. Besides, we provide more detailed information for reference:
>
> - **Lacking effective annotation tool**: As we know, there is no one open-source video text annotation tool. And video annotation tools for the general objects, *e.g., VOTT (https://github.com/microsoft/VoTT#using-vott), UltimateLabeling (https://github.com/alexandre01/UltimateLabeling)*, show inefficient annotation method, and do not support labeling rotated box annotation and recognition results. We build a private efficient tagging tool for video text annotation.
> - **The labor-intensive job.** The labeling process of 1,010,848 video frames on MMVText takes 30 men in two months, i.e., 20,160 man-hours. Most university labs cannot afford the huge annotation cost.
> - **The requirement from Video Tasks**. In the past, most researchers focus on image-based text spotting due to the immaturity of the technology and less requirement for video-level application. Recently, with the great development of image-based text spotting and huge requirement from video applications, increasing researcher and many international corporations, *e.g., YouTube, Tik Tok, KuaiShou*, eager to develop video-based text spotting for favoring many video-based downstream tasks (video retrieval, video understanding).
>
> **Q5:** *No video-based detection/recognition methods for experiments.* **A5:** Sorry about this, but we still want to further provide some detailed information for reference:
> - **No open-source and complex pipeline.** There are few video-based methods and **no one is open source**. Besides, most exiting video-based methods are proposed before 2019, their pipeline is extremely **complicated**, difficult to re-implement. Cheng et al. [2] proposed a video text spotting method, which includes *four modules*: spatial-temporal detector, text tracking, quality scoring, recognition. Without reference and more details, it's hard to reproduce the model and performance.
> - **The convincing experiments.** Similar, most existing datasets, *e.g., RoadText-1k[1],* do not provide a video-based method for experiments. The experiments using image-based detection and recognition track text instances temporally based on attributes of the text objects in multiple frames, which also evaluate video text tasks efficiently.
>
> **Q6:** *What makes some of the video types (e.g. Campus, Travel, Celebrity) more difficult than others?* **A6:**  We will replenish more detailed insights in the Camera Ready. Actually, according to the visualization and analysis ([YouTube]( https://www.youtube.com/watch?v=_Yykx0qgJZ4)( https://www.youtube.com/watch?v=_Yykx0qgJZ4)), the low performance on these difficult video types (*e.g., Campus, Travel, Celebrity*) is caused by text blur, out-of-focus, a new theme for the scenario, and other factors. Outdoor scenes video (*e.g., Campus, Travel, Celebrity*) usually have a **more complex environment and unsteady camera movement**. On the contrary, some easy video types such as cartoons hardly contain complex scene text, and the camera movement extremely stable by manual handling.
>
> **Q7:** *Fig 3 (c) and Table 2 could have included a more detailed language breakdown, e.g. include English specifically.*  **A7:** Great suggestion! We will add English as a new class to Fig 3 (c) and Table 2 in the revised version, as it would help to further evaluate multilingual recognition. The updated Figure and Table can be found in https://github.com/weijiawu/MMVText-Benchmark/blob/main/Dataset/image/RecognitionLanguages.png.
>
> **Q8:** *We do not know how good the quality of the interpolation is?*  **A8:** As the reviewer mentioned, the quality of the interpolation is not satisfactory in some cases (*e.g., new text appears, or text disappears*). To address the issue, we manually screen and re-label them. Around 150,000 video frames from 1,010,848 video frames are selected to refine, taking 10 men in two weeks. We will make it clear in the Camera Ready, as shown in the screenshot(https://github.com/weijiawu/MMVText-Benchmark/blob/main/Dataset/image/quality.png). To further show the annotation quality, we sample and visualizing several scenarios videos randomly in  https://www.youtube.com/watch?v=WrQER3-AHsA (1080p high resolution).
>
>
> [1] Roadtext-1k: Text detection & recognition dataset for driving videos. In IEEE International Conference on Robotics and Automation,  2020.
>
> [2] You only recognize once: Towards fast video text spotting. In ACM International Conference on Multimedia, 2019.
>
> [3] Mot20: A benchmark for multi-object tracking in crowded scenes. arXiv preprint arXiv:2003.09003 (2020).
>
> [4] Tracking Pedestrian Heads in Dense Crowd. In CVPR. 2021.

---

> ### Author Response · Authors · 2021-07-12
> **Response to Reviewer qYEt (Part3)**
>
> **Q9:** *It is unclear how the "time matching, space location matching, and recognition result matching" are defined?*  **A9:** In frame $t$, $h_t$ and $o_t$ denote hypothesis set (predicted IDs $I_p$ , box locations $L_p$, recognition results $R_p$ ) and ground truth set (IDs $I_g$, box locations $L_g$, recognition ground true $R_g$ ). We define "time matching", "space location matching" and "recognition result matching" as follows:
> - Time Matching： $I_p⟺I_g$
> - Space Location Matching: $IoU(L_p,L_g)>0 $
> - Recognition Result Matching: $R_p= R_g$
>
> where $⟺$ denotes both are the same identity. A correct match (true positive) should fit the above three conditions. In fact, recognition match is more difficult than the location match (location iou>0) since we argue that the final recognition results are more important for other downstream tasks. The evaluation code has been released at https://github.com/weijiawu/MMVText-Benchmark/blob/main/Evaluation_Protocol/Task2_VideoTextSpotting/evaluation.py, and we will make it clear in the Camera Ready.
>
> **Q10:** *Fig 3 (b) is supposed to show two methods (PSENet and CRNN) but only one plot is present.* **A10:**
> Sorry to **confuse** you. Fig 3(b) shows the end-to-end video text spotting performance, which needs two modules: text detection task (PSENet) and recognition task (CRNN). With the detection and recognition results, we link and match text objects in the current frame and next frames by IOU and edit distance of text by Text Tracking Trajectory Generation, Line 249-254. We will make it clear in the revised version.
>
> **Q11:** *A very good motivation for the new benchmark.* **A11:** Our motivation mainly comes from the following two points:
> - Most existing video text datasets are proposed before 2019 years, which cannot support large-scale caption text, various scenarios, and multilingual detection/recognition tasks. Besides, some new challenges and insights need to focus on, please refer **to Q2/A2 for details**.
> - There is an urgent need for large-scale video text information to develop language-and-video research tasks, *e.g., CLIPBERT[5], Video Retrieval[6][7], and Video Captioning[8]*. However,  the existing dataset cannot meet the requirement due to the small scale(<10,000 frames) and single scenario for large-scale (>1,000,000) language-and-video tasks[5][6][7][8].
>
> **Q12:** *Are there established terms for how different tasks in this work are typically referred to.* **A12:**
> Sorry to confuse you. We will define the standardized terms for the four tasks: Text Detection, Text Recognition, Video Text Tracking, and End to End Video Text Spotting in the Camera Ready.
>
> **Q13:** *L30-36: The authors argue for challenges of text detection/recognition in video as opposed to images, are there any references to cite to support this discussion?* **A13:** The previous work, *e.g., RoadText-1K[1] and LSVTD[2]*, all provide **relevant cases and analysis** to support the discussion. The main challenges of text detection/recognition in the video are various factors from video like blur, out-of-focus, other artifacts/distortions, and unsteady camera movement. Besides, spatio-temporal information cannot be obtained by image-based methods, which cannot meet the requirement from some downstream tasks such as video caption.  We will make it clear in the Camera Ready.
>
> **Q14:** *L43 according to this, YouTube Video Text (YVT) also has two types of text annotations, captions, and scene text. However, in Table 1 it is shown as only "Scene" text?* **A14:** Thank you for the comment, we will fix the error in the Camera Ready, as shown in https://github.com/weijiawu/MMVText-Benchmark/blob/main/Dataset/image/Statistical%20Comparison1.png
>
> **Q15:** *L54 "This makes it difficult to evaluate the effectiveness of more advanced deep learning models." - why?* **A15:** The attributes (*e.g., small scale, single language, and single scenario*) of the existing dataset cannot meet the requirement for some scenario algorithm evaluations. For more details please refer to **Q2/A2** for contribution details and experiment insights. We will adjust the statement to “This makes it difficult to evaluate the effectiveness of more advanced deep learning models for some scenarios” for a more well-knit statement in the revised version.
>
> **Q16:** *L64-66 odd to refer to the dataset as "multidimensional", not sure what the authors mean?* **A16:** Sorry to confuse you. We want to present the various scenarios of MMVText by the term "multidimensional". We will fix these confusing error and polishing the manuscript by a native speaker or polish institutions, e.g., American Journal Experts (https://www.aje.com/?_gl=1*1rq17qp*_gcl_aw*R0NMLjE2MjYwODgyMTUuRUFJYUlRb2JDaE1JbmEycHpiTGQ4UUlWY3R4TUFoMGJDd0hrRUFBWUFTQUFFZ0txdmZEX0J3RQ..&_ga=2.147751092.2064943953.1626088215-1440899898.1615044050&_gac=1.237637172.1626088215.EAIaIQobChMIna2pzbLd8QIVctxMAh0bCwHkEAAYASAAEgKqvfD_BwE)

---

> ### Author Response · Authors · 2021-07-12
> **Response to Reviewer qYEt (Part4)**
>
>
> **Q17:** *Table 1: What is the difference between "Incidental" and "Open" scenario? I think both are open, so perhaps include #categories?* **A17:** We define **indoor and outdoor** scenarios in daily life (*e.g., walking outdoors, driving, searching for a shop in a shopping street, browsing products in a supermarket*) by the term `''Incidental'' without others *e.g., news report, virtual world, game, NBA*. And the term “Open” denotes **any scenarios**, includes *Game (PUBG mobile, Arena of Valor...), Sport(NBA, FIFA World Cup...), News Report, TV program, Education(Campus, Classroom, Book, Lesson), Technology(Introductory video, Scientific Propaganda), and so on*. We will make it clear in the Camera Ready.
>
> **Q18:** *L 195, 203 Is "End to end video text spotting task' an existing task or a new task? Initially it seems to be an existing task, hence the authors discuss that it is "supported" by other datasets, but then they introduce a new metric specific to the task, suggesting it is new?* **A18:** End to end video text spotting task is an existing task, but the Identification $F1$($ID_{F1}$) from Multiple Object Tracking Task[9][10], as a **new metric**, is adopted firstly for video text spotting. More importantly, we modify the $ID_{F1}$ to $TID_{F1}$, which focuses on the recognition results that be required by many downstream tasks. Therefore, we provide much detailed information for the metric.
>
> **Q19:** *L147 what is "video frame recovery"?* **A19:**
> Thank you for the valuable comment. The term refers to the **how to linear interpolation** for unlabeled video frames L 139, 152. We have fixed the confusing error in the revised version, as shown in the screenshot(https://github.com/weijiawu/MMVText-Benchmark/blob/main/Dataset/image/interpolation%20.png).
>
> **Q20:** *L154 what is "double detection correction"?* **A20:**
> We want to state that an audit team is invited to carry out another round of annotation checks. We have fixed the confusing error in the revised version, as shown in the screenshot(https://github.com/weijiawu/MMVText-Benchmark/blob/main/Dataset/image/correction.png).
>
> **Q21:** *Some additional references may be the new Vision&Language tasks that include scene text: [a] Singh, Amanpreet, et al. "Towards vqa models that can read." Proceedings of CVPR. 2019. [b] Sidorov, Oleksii, et al. "Textcaps: a dataset for image captioning with reading comprehension." ECCV. Springer, Cham, 2020.* **A21:** Thank you for the valuable suggestion. Both are great work for Vision&Language tasks, especially for TextCaps, which provides a wonderful benchmark with 145k captions and 28k images for the image captioning task. We will add them both as references in the Camera Ready, as shown in the screenshot(https://github.com/weijiawu/MMVText-Benchmark/blob/main/Dataset/image/TextCaps.png).
>
> **Q22:** *The authors mention a survey for video w/ text methods [41] L82, however, they only discuss two methods, [36] and [4]. I assume there are more than two methods in the survey?* **A22:**
> Thank you for the valuable suggestion again. We will supplement and introduce more related video-based tracking methods[11][12] in the revised version, as shown in the screenshot(https://github.com/weijiawu/MMVText-Benchmark/blob/main/Dataset/image/relatedwork.png)
>
> **Q23:** *To the checklist question "Did you discuss any potential negative social impacts of your work?" the answer was given No. The authors could think about it and add some comments to the conclusion/discussion section in the future revision.* **A23:**  Thank you. We have supplemented the discussion in the revised version, as shown in the screenshot(https://github.com/weijiawu/MMVText-Benchmark/blob/main/Dataset/image/Negative.png)
>
> **Q24:** *Sec 3.5 seems out of the place where it is. Could go to conclusion/discussion?* **A24:** Thank you, the suggestion has been taken in the revised version, as shown in the screenshot(https://github.com/weijiawu/MMVText-Benchmark/blob/main/Dataset/image/place1.png).
>
>
> [5] Less is more: Clipbert for video-and-language learning via sparse sampling. In CVPR. 2021.
>
> [6] Tvr: A large-scale dataset for video-subtitle moment retrieval. In CVPR. 2020.
>
> [7] Msr-vtt: A large video description dataset for bridging video and language. In CVPR. 2016.
>
> [8] Vatex: A large-scale, high-quality multilingual dataset for video-and-language research. In ICCV. 2019.
>
> [9] Cvpr19 tracking and detection challenge: How crowded can it get? arXiv preprint arXiv:1906.04567, 2019.
>
> [10]  Performance measures and a data set for multi-target, multi-camera tracking. In Workshops of ECCV. 2016.
>
> [11] Video text detection and recognition: Dataset and benchmark, in Proc. IEEEWinter Conf. Appl. Comput. Vis., Mar. 2014.
>
> [12] Scene text recognition in multiple frames based on text tracking,” in Proc. IEEE Int. Conf. Multimedia Expo (ICME), Jul. 2014.

---

### Official Review · Reviewer_NPLq · 2021-07-04
**A dataset for video text spotting**

**Rating:** 4
**Confidence:** 3
**Correctness:** Yes.
**Clarity:** The presentation is clear.

**Strengths:**

1. The dataset seems to contains the largest number of frames when compared with the existing benchmarks.
2. Video text spotting is crucial for many applications in the real world.
3. Much labor effort is needed to collect the data annotation.

**Weaknesses:**

The reviewer has a lot of concerns about the paper.
1. The dataset has a similar size of the existing benchmarks and does not show enough evidences that the new benchmark is different from the existing dataset or has unique important features that other existing datasets can not provide.
2. Although the dataset seems to have the largest number of frames on video text spotting, the dataset seems to have smaller video number than existing datasets like [26]. How to make sure that the video texts in MMVT is more diverse than existing datasets since a video may contain many similar and duplicated frames. How many unique text instances are provided in MMVT and other existing datasets?
3. The reviewer suggests the authors use a table to summarize the statistics of the MMVT and existing datasets, which will make the dataset comparison more clear and easier to distinguish.
4. The dataset is divided into two categories and claim that categories can favor other tasks such as video understanding, video retrieval and video text translation. However,  no experiments in Section 4 support the claim.
5. Experiments do not show much new insights about existing methods or the uniqueness of the new benchmark compared with the existing benchmark;
6. Figure 3 (a)-(b) are too small to read on the paper.
7. The current version of the paper still contains many typos or grammar  mistakes, e.g. Line 292 and Line 307. A proofreading is needed.

**Additional Feedback:**

None.

**Documentation:**

Yes.

**Ethics:**

No.

**Relation To Prior Work:**

Please see the "weaknesses" section.

**Summary And Contributions:**

This paper proposes a new dataset for video text spotting. The dataset contains 510 videos with 1000K frames with 30 categories. It separates texts in the videos into caption text and scene text. Experiments are conducted and analyzed on the benchmark, showing the performance of existing baselines for text detection, text recognition and video text tracking.

---

> ### Author Response · Authors · 2021-07-09
> **Response to Reviewer NPLq (Part1)**
>
> Thank the reviewer for the valuable comments and suggestions.
>
> **Q1:** *The dataset is similar to the existing benchmarks.*  **A1:** We want to further state the contributions of MMVText:
> - **Size**. MMVText is **15 times** larger than the existing largest dataset (LSVTD[1]) with various scenarios(1,010,848 *v.s*  66,700 frames). RoadText1k[2] contains 300k video frames, but it only supports driving scenarios.
> - **Supported Language.** As we know, MMVText is the first large-scale multilingual video text benchmark.
> - **Video Scenarios.** MMVText provides various open scenarios, including many **new scenarios** such as Sports(NBA, FIFA World Cup...), Vlog, Game, etc. To further present the diversity of MMVText, we summarized detailed statistics in https://github.com/weijiawu/MMVText-Benchmark/blob/main/Dataset/image/Statistical%20Comparison1.png.
> - **Annotation.** Caption(*e.g.,* title, logos) and scene text(*e.g.,* street signs) are separately tagged for the two different meanings. Besides, MMVText supports rotate box annotation(four points), but many existing datasets(RoadText-1k, YVT) only support bounding box annotation(two points).
> - **Data Quality.** Videos in MMVText support high resolution (1080p), but most existing datasets(RoadText-1k, YVT)  only support 720p.
> **Conclusion.** Most existing datasets are proposed before **2019 years**, which can not meet the requirement for evaluation due to the small size and low-quality annotation. The download link of YVT[5] even has become invalid. MMVText, as a standardized benchmark, would facilitate video-and-language research.
>
> **Q2:** *Comparison with RoadText-1k[2], why MMVText is more diverse?*  **A2:** We summarized more detailed statistics concerning the diversity of MMVText in https://github.com/weijiawu/MMVText-Benchmark/blob/main/Dataset/image/Statistical%20Comparison1.png. RoadText-1k are sampled from driving datasets BDD100K[4], which **only provides driving scenarios**. MMVText includes 30 open scenarios(e.g., Driving, Vlog) selected from online resources that are entirely different. We want to further emphasize that many scenarios, *e.g., Game(PUBG mobile…), Sport(NBA, FIFA World Cup…), News Report, TV...*, which are **not supported** by the existing datasets.  To further show the diversity of MMVText, we sample and visualizing several scenarios videos randomly in https://www.youtube.com/watch?v=WrQER3-AHsA (1080p high resolution).
>
> **Q3:** *Dataset statistics comparison using a table?* **A3:** We have been provided a table about the comparison in Table. 1. We guess that the suggestion is about showing more detailed information, *e.g., how many videos/frames there are for each of the languages?* We will add it in the Camera Ready:
> - **The data distribution for 30 open scenarios** (https://github.com/weijiawu/MMVText-Benchmark/blob/main/Dataset/image/Data%20Distribution.png)
> - **The data distribution for text language and category** (https://github.com/weijiawu/MMVText-Benchmark/blob/main/Dataset/image/languageAndCategory1.png)
>
> **Q4**: *No experiment supports the benefit for other tasks such as video retrieval?* **A4:**  We provide examples, analysis, and references in A.2, Supplementary Material. Although the annotation of text has been provided, for the video retrieval task, we still need to define queries and retrieved results annotation, as well as a new metric and baseline. Due to **the huge amount of work**, we can not provide the related experiments in this version. More video-and-language tasks will be supported in our future works(e.g., video retrieval[3]).
>
>
> [1] You Only Recognize Once: Towards fast video text spotting. ACMM, 2019
>
> [2] Roadtext-1k: Text detection & recognition dataset for driving videos. ICRA, 2020
>
> [3] Tvr: A large-scale dataset for video-subtitle moment retrieval. ECCV, 2020
>
> [4] BDD100K: A diverse driving video database with scalable annotation tooling, CoRR, 2018
>
> [5] Phuc Xuan Nguyen, Kai Wang, and Serge Belongie. Video text detection and recognition:Dataset and benchmark. In IEEE winter conference on applications of computer vision, pages 776–783, 2014.

---

> ### Author Response · Authors · 2021-07-11
> **Response to Reviewer NPLq (Part2)**
>
> **Q5:** *Experiments do not show new insights.* **A5:** We want to highlight three new insights from our experiments:
> - **New Scenarios, New Challenge.** The proposed new scenarios, *e.g., Celebrity, Campus, Dance*, present **unsatisfactory** results than others (*e.g., Cartoon*), as shown in Figure 3 (a,b) and Attribute Experiments (Line 280-284). The low performance is mainly caused by blur, out-of-focus of text, and complex themes in the new scenarios.  The comparison for Cartoon and Celebrity can be found in https://www.youtube.com/watch?v=_Yykx0qgJZ4. (Green and blue boxes denote annotation and prediction of PSENet, respectively. ) Outdoor scenes video (e.g., Campus, Travel, Celebrity) usually have a more **complex environment** and **unsteady** camera movement. On the contrary, some easy video types such as cartoons hardly contain complex scene text, and the camera movement extremely stable by **manual handling**.
> - **Multilingual Recognition.** Unlike English, Chinese contains thousands of characters, which are difficult to recognize (**3,856 Chinese characters v.s. 26 English characters on MMVText**). Table 2 and Figure 3 (c) in Attribute Experiments prove the conclusion. We also present some bad cases in https://github.com/weijiawu/MMVText-Benchmark/blob/main/Dataset/image/recognition1.png.  Many **clear** image patches with Chinese text still bring recognition errors, since the classification of such many characters is **difficult** for the network to learn. By comparison, most English errors are caused by blur and out-of-focus. MMVText, as *the first large-scale multilingual dataset*, **15 times** larger than LSVTD, can promote multilingual recognition.
> - **Challenge from Caption Text.** MMVText, *as the first large-scale caption dataset*, provides the detection results for caption and scene text in Figure 3 (d), which shows the result of scene text is better than the counterpart of the caption. The prime reason is that captions usually are long text (**average width-height ratio: 6.8 for caption *v.s.* 2.3 for scene text**), the difficult cases to detect. The bad case of the caption can be found in https://www.youtube.com/watch?v=MhPuxW_bdos. (green and blue boxes denote annotation and prediction of EAST, respectively).
>
> **Q6:** *Figure 3 (a)-(b) are too small to read on the paper.* **A6:** Thank the reviewer again for the valuable suggestions. In Figure 3 (a)-(b), we want to highlight that some of the new video types, *e.g., Campus, Dance, Celebrity*, show unsatisfactory performance than others. These scenarios **cannot be supported** by the existing datasets.  We will refine it in the Camera Ready.
>
> [1] You Only Recognize Once: Towards fast video text spotting. ACMM, 2019
>
> [2] Roadtext-1k: Text detection & recognition dataset for driving videos. ICRA, 2020
>
> [3] Tvr: A large-scale dataset for video-subtitle moment retrieval. ECCV, 2020
>
> [4] BDD100K: A diverse driving video database with scalable annotation tooling, CoRR, 2018
>
> [5] Phuc Xuan Nguyen, Kai Wang, and Serge Belongie. Video text detection and recognition:Dataset and benchmark. In IEEE winter conference on applications of computer vision, pages 776–783, 2014.

---

### Official Review · Reviewer_ZKz4 · 2021-07-05
**review of MMVText**

**Rating:** 7
**Confidence:** 4
**Clarity:** Yes. The paper is well written.

**Strengths:**

+ The dataset is larger and more diverse than existing datasets in terms of number of frames and number of text instances.

+ The scenarios that the dataset contains are diverse and open-world, which poses more challenges to research on the community text spotting.

+ The dataset annotates both caption and scene text in videos.

+ The authors have clearly stated the evaluation metrics and open-sourced the evaluation protocols.

**Weaknesses:**

- Since the dataset is collected from Kuaishou, are the videos easy to access and free of copyright issue?

- Ethical issues: From example images in Figure 2, human faces are observed in multiple frames. Is there any potential risk to use them? Preprocessing such as blurring human faces may be considered.



====after rebuttal====

- The copyright issue is resolved according to the authors' response.

- A stronger blurring is better for privacy preserving.

**Additional Feedback:**

No.

**Correctness:**

Mostly correct. The data collection seems reasonable and the evaluation metrics are clearly stated.

- I have a particular question in terms of the experiments on multilingual text recognition. For the setting of "CH/AN", do you train a single network and combine both Chinese characters and alphanumeric characters and only evaluate separately or do you train two networks separately? This may help answer whether the multilingual property or the blur quality contributes more to the low accuracy.


====after rebuttal====

This question has been answered. A ultimate goal would be to use a single model to recognize bilingual or even multilingual characters. Though this setting is much more challenging and the overall accuracy would be even lower, this may make the recognition pipeline much simpler. Otherwise you would need two separate networks and an additional module to decide which language network to switch to. The authors can consider discussing these in the final version.

**Documentation:**

The authors have provided a GitHub repository to host the evaluation protocols.

**Ethics:**

From example images in Figure 2, human faces are observed in multiple frames. Is there any potential risk to use them? Preprocessing such as blurring human faces may be considered. Also see weakness 2.

**Relation To Prior Work:**

Yes. The paper clearly discusses the relation to prior works.

**Summary And Contributions:**

The paper proposes a large-scale dataset for video text spotting. The dataset covers a diverse range of categories and provides multilingual text annotation for both scene text and caption text. It also establishes evaluation protocols and conducts comprehensive experimental results on tasks including text detection, recognition, tracking and end-to-end spotting.

---

> ### Author Response · Authors · 2021-07-10
> **Author response to the review based on the feedback**
>
> Thank the reviewer for the valuable comments and suggestions.
>
> **Q1**: *Since the dataset is collected from Kuaishou, are the videos easy to access and free of copyright issue?*.
>   **A1:** The dataset has been approved by the legal department, Kuaishou Technology. Without copyright issues, the dataset can be used freely for research.
>
> **Q2**:  *Ethical issues: From example images in Figure 2, human faces are observed in multiple frames. Is there any potential risk to use them? Preprocessing such as blurring human faces may be considered.*
>   **A2:** Thanks for the valuable suggestion. Actually, blurring human faces slightly has been adopted for Figure 2. To further avoid potential risk, we will make a greater degree of obscuring. Besides, **similar to the demo video** in supplementary materials or new demo in https://www.youtube.com/watch?v=WrQER3-AHsA (supporting1080p high resolution), obscuring processing has been taken for all videos used on MMVText.
>
> **Q3**: *Multilingual text recognition. I  have a particular question in terms of the experiments on multilingual text recognition. For the setting of "CH/AN", do you train a single network and combine both Chinese characters and alphanumeric characters and only evaluate separately or do you train two networks separately? This may help answer whether the multilingual property or the blur quality contributes more to the low accuracy.*
>   **A3:** Sorry to confuse you. We train two same recognition networks on Chinese characters and alphanumeric characters, **separately**. The setting can help to avoid mutual interference from different languages just as you indicated, other factors(*e.g., blur quality, character number*) may contribute more to the low accuracy. Unlike English(26 characters), Chinese contains thousands of characters(**3,856 Chinese characters v.s. 26 English characters on MMVText**), which are difficult to recognize. We also present some bad cases(https://github.com/weijiawu/MMVText-Benchmark/blob/main/Dataset/image/recognition1.png) concerning multilingual text recognition to support some insights:
> - **3,856 Chinese characters bring a new challenge**. From the visualization, we argue that many clear image patches with Chinese text still bring recognition errors, since the classification of such many characters is difficult for the network to learn. By comparison, most English errors are caused by blur and out-of-focus.
> - **Long caption text brings a new challenge**.  We argue that long caption text is more difficult to detect ([YouTube demo](https://www.youtube.com/watch?v=MhPuxW_bdos)) and recognize than scene text since there are more characters and an extreme width-height ratio(*average width-height ratio: 6.8 for caption v.s. 2.3 for scene text on MMVText*).  But no existing datasets support the caption text, except an outdated dataset YVT[1] that was proposed in 2014, and the download link even has become invalid.
>
>
>
> Thank you for the recognition. MMVText, as a practical standardized benchmark, would facilitate video-and-language research for the community. Besides, more [statistics comparison](https://github.com/weijiawu/MMVText-Benchmark/blob/main/Dataset/image/Statistical%20Comparison1.png) and [statistics details](https://github.com/weijiawu/MMVText-Benchmark/blob/main/Dataset/image/languageAndCategory1.png) will be included to further refine our paper in the Camera Ready.
>
> [1] Phuc Xuan Nguyen, Kai Wang, and Serge Belongie. Video text detection and recognition: Dataset and benchmark. In IEEE winter conference on applications of computer vision, pages 776–783, 2014.

---

### Decision · Program_Chairs · 2021-07-26

**Decision:**

Reject

**Comment:**

The paper introduces a new large-scale dataset for text spotting called MMVText which consists of 510 videos where all text instances have been annotated with bounding boxes, track IDs, transcription, and classification labels (scene text / caption). In total the dataset contains more than 1M labeled images and 4.5M labeled text instances.

Initially reviewers were somewhat negative on the submission, and raised a number of issues about the dataset, experiments, and clarity of presentation. The authors provided comprehensive responses that addressed many of the issues initially raised by reviewers; in the end reviewers were split, with Reviewer ZKz4 advocating the most strongly for acceptance, Reviewer qYEt marginally in favor of acceptance, and Reviewer NPLq advocating rejection.

The dataset itself is impressive. The scale, variety, and multilingual nature of the dataset are a clear improvement over similar existing datasets for this task. This dataset is likely to be useful and impactful within the subfield of text spotting.

However, the AC agrees with some of the weaknesses raised by Reviewers qYEt and NPLq. The most salient issue is that the paper does not empirically demonstrate the utility of the proposed dataset compared to existing datasets. The largest prior dataset (RoadText-1K) has more videos (1000 vs 510) but fewer frames (300k vs 1M), and covers a different data distribution (monolingual driving videos vs multilingual open-domain videos). The most similar prior dataset (LSVTD) is multilingual and contains videos of different scenarios, but annotates only scene text and is much smaller (100 videos vs 510, 67k frames vs 1M). Given the qualitative differences between MMVText and prior datasets, the authors should have provided a more convincing empirical comparison to demonstrate the quantitative benefits to MMVText and disentangle the utility of the various qualitative differences. For example, how well do models trained on RoadText-1k, or LSVTD, or the union of the two, perform when evaluated on MMVText? If they perform worse than models trained on MMVText, is the performance gap due to dataset size, or shift in scene distribution, or other factors? Controlled experiments could help provide insight and better quantify the utility of this new dataset.

In addition, Reviewers ZKz4 and qYEt raise some ethical questions about the dataset that have not been fully addressed. In particular, the authors “have blurred identifiable information or offensive content”, but do not provide any details about how this was achieved. The provided demo video (​​https://www.youtube.com/watch?v=WrQER3-AHsA) contains many instances of unblurred faces, which does not match the author’s response to this question. In addition, the initial submission did not discuss any potential negative societal impacts of the work; the author response amended the paper to include a one-sentence discussion mentioning “potential negative societal impacts for personal privacy”. This should be elaborated, as the proposed discussion does not suggest the authors have given much thought to the question of potential societal impact of their work.

Due to these issues, the AC feels that the paper is not ready for publication in its current form. However we emphasize that the dataset itself is novel and interesting, and is the type of contribution we want to encourage in this track. Therefore we strongly encourage the authors to take all feedback from reviewers into consideration, revise the paper, and resubmit to the second round of the Datasets and Benchmarks Track in August.